# Pathogenesis of Cerebral Malaria: New Trends and Insights for Developing Adjunctive Therapies

**DOI:** 10.3390/pathogens12040522

**Published:** 2023-03-27

**Authors:** Praveen Kishore Sahu, Sanjib Mohanty

**Affiliations:** 1Department of Molecular Biology & Infectious Diseases, Community Welfare Society Hospital, Rourkela 769042, Odisha, India; 2Infectious Disease Biology Unit, Ispat General Hospital, Rourkela 769005, Odisha, India

**Keywords:** cerebral malaria, pathogenesis, *Plasmodium falciparum*, brain swelling, adjunctive therapies

## Abstract

No specific or adjunctive therapies exist to treat cerebral malaria (CM) as of date. CM is a neuropathological manifestation of the malaria infection in humans, caused by the hemoparasitic pathogen *Plasmodium falciparum*. Driven through a multitude of virulence factors, varied immune responses, variations in brain swelling with regard to the age of patients, parasite biomass, and parasite-typing, the essential pathogenetic mechanisms underlying clinical CM have remained elusive. However, a recent series of studies based on molecular, immunologic, and advanced neuroradiologic and machine-learning approaches have unraveled new trends and insights to better understand and focus on the key determinants of CM in humans. This could possibly be the beginning of the design of new and effective adjunctive therapies that may not be common or applicable to the entire malarious world, but that could, rather, be specific to the variations in the determinants of CM.

Malaria is a major public health menace in the tropical and sub-tropical regions of the world, with an estimation of 247 million infections and 619,000 human deaths occurring from severe complications in 2021 [1]. Cerebral malaria (CM) is a serious, life-threatening neurological complication of malaria, caused predominantly by the hemoparasitic pathogen *Plasmodium falciparum* amongst the other species of *Plasmodia* that infect humans. The inability to treat cerebral malaria in humans is primarily due to the lack of a specific anti-CM drug or an adjuvant. Considering the multifactorial nature of the neuropathology in cerebral malaria, even the most potent antimalarials [2,3] fail to treat or restrict the development of cerebral malaria, especially in cases with delayed diagnosis or due to difficulties in accessing adequate healthcare in resource-poor settings. The quest for the development of novel and effective adjunctive therapies has thus become one of the prime foci of severe and cerebral malaria research.

Cerebral malaria is the neuropathological manifestation of malaria, effectively constituted through contributions from the parasite and the human host [3,4]. The pathology of the brain is manifest through the collective occurrence of several host and parasitic processes at the pathogenic, immunologic, and neurologic levels, involving a complex interplay between various factors and a multitude of molecules resulting in the ‘clinical syndrome’ of CM. The clinical presentation of CM is diverse in adults and children. The spectrum spans from irritability and altered sensorium to moderate or deep coma and, less frequently, to seizures (in adults more so than in children) [5]. When associated with other organs failure(s), e.g., kidney, liver and lungs, CM results in associated mortality reports as high as 43% and 59% in adults and children from South-East Asia and India, respectively [2,6], with most cases (61%) dying within the first day of hospitalization [6]. Additionally, development of post-recovery neuro-cognitive impairment is frequent in children with CM [7,8], which is often life-changing.

The detailed pathophysiology leading to the development of CM and brain injury remains an enigma. However, major postulations pointed to the sequestration theory that cytoadherence of the parasitized red blood cells (pRBCs) on to the endothelial cells, facilitated by rosette formation in the cerebral microcapillaries, causes brain ischemia, and the resulting immune-inflammatory responses lead to the concurrent onslaught on the blood-brain-barrier (BBB) by a cytokine storm and reactive oxygen species [9,10,11,12,13,14]. The breach in the BBB causes brain swelling, which manifests as an encephalopathy or cerebral syndrome in the patients with CM. Nonetheless, the major limitation of these postulations was that the majority of inferences were drawn from either an experimental model of cerebral malaria or else autopsy-based investigations [15,16]. Therapeutic potential of osmotic diuretic drugs like mannitol, which reduces cerebral edema, was tried in CM patients as an adjuvant with no beneficial effects [17,18]. Interestingly, researchers started turning to radiological evaluations of the brain using computed tomography (CT) for mapping cerebral changes or swelling in CM patients [18]. However, the possible mechanism underlying the observed swelling of the brain of varying degrees remained elusive until the advent of more precise and even high-resolution neuroimaging technologies.

A series of recently demonstrated findings on CM patients from India and Africa using magnetic resonance imaging (MRI) based neuro-radiological studies started shedding light on several grey areas and widened the horizon to better understand the key pathogenetic mechanisms of action during the development of CM. For example, in a subset of non-fatal CM patients from India, following standard treatment, the increase in brain volumes was found to be caused by vascular engorgement and reversible impairment of BBB through posterior reversible encephalopathy syndrome—PRES [19]. Consistent with this, a study from Zambia, southern Africa, demonstrated that PRES was associated with a reversible coma in patients with CM without any gross impairment to the BBB or focal cortical abnormalities evident from MR imaging findings [20]. Subsequently, there were distinct pathogenic patterns identified in Indian adult and children with fatal CM, which was characterized by severe brain swelling in children and generalized brain hypoxia in adults, respectively, through apparent diffusion co-efficient (ADC) maps and measurement of elevated plasma levels of lipocalin-2 and miR-150 [21]. The cause of death, however, in Indian children mimicked the African cohort of children from Malawi, which had revealed massive brain swelling during CM and herniation of the brainstem upon MR imaging [22]. Subsequently, following an in-depth comparison of the Indian cohort (adult and children) and the African cohorts of children with CM based on the molecular, immunological, and neurological profiles aided by machine-learning models, it was demonstrated that the pathogenetic mechanisms of CM may have common brain swelling determinants, e.g., elevated levels of Endothelial protein C receptor (EPCR) binding *var* gene transcripts as well as higher levels of parasite biomass [23]. In this study, the role of parasite biomass as a pivotal pathogenic mechanism in conjunction with a higher expression of *var* gene transcripts (EPCR binders) in CM patients from Rourkela, eastern India, was found to be consistent with an earlier study on severe adult malaria patients from Goa, western India [24]. Subsequently, it was revealed that there could be coma-independent triggers from severe malaria in adults, e.g., creatinine and S100B (plasma biomarkers for acute kidney and neural injury, respectively), which can lead to cytotoxic brain changes apart from the aggravation from higher parasite biomass, causing local brain hypoxia [25].

In the past few decades, an array of tools, models, and biomarkers have been identified to better understand the pathogenesis of severe and cerebral malaria, and many candidate adjuvants have been proposed with different mechanisms of action and promising the potential to treat CM [26]. Despite the demonstration of huge success in experimental or animal models, clinical trials of numerous adjunctive therapies targeted for human CM in the past have yielded little or no success in humans, as extensively reviewed earlier [27]. The recent series of studies on CM pathogenesis using neuroradioimaging has shown the distinction between the underlying pathogenetic mechanisms of CM and the type of determinants related to the clinical presentation of CM along with other organ association. These may include but are not limited to the levels of parasite biomass [24], *var* transcripts-binding phenotypes [28], varying degrees of brain swelling [19,21,22], and, more recently, the silent confounders from severe non-cerebral malaria such as cross-talk between the brain and other organs like the kidney [25]. Therefore, the quest and the role of emerging and effective adjuvants and novel therapeutic approaches will become increasingly important, and should be monitored very closely until there is a ‘proper’ adjunctive therapy for CM. Amongst others, for example, the role of glutamate antagonists for the reversal of both cytotoxic edema and BBB disruption [29] and promoting the balance of TH1/TH2 response and the upregulation of neurotrophic factor levels in the frontal cortex and hippocampus [30] respectively, and whole blood transfusion therapy for the prevention of cerebral ischemia and aggravation of BBB breakdown [31] in experimental models of CM have exhibited promising potential, but these studies deserve evaluation in clinical CM. In addition, the early benefits of bone marrow-derived mesenchymal stromal cells on cerebral malaria pathologies observed in the experimental mouse model [32] and the overall effects of stem-cell therapies must be evaluated carefully in non-human primate models before extrapolation to human CM [33]. Besides, clinical trials are already in progress to explore the merits of mechanical ventilation and treatment with intravenous hypertonic saline in African children with CM, with the aim to reduce brain swelling [34]. These new approaches along with the future studies directed towards new discoveries provide good reasons to raise hope and count on forthcoming findings to devise therapeutic options for CM, but there should still be careful considerations.

In view of the recent trends and the new insights gained from molecular, immunological, and neuroradiological approaches to understand CM pathogenesis, it is now evident that a common adjunctive therapeutic for CM will not be effective across the whole of the malarious world, and especially for adults and children [35]. Moreover, the mounting evidence from fluorescein angiography studies focused on the retina during CM strongly suggest the rationale to develop separate adjunctive interventions to address ischemic injury and brain swelling, respectively [36]. It is well understood that the activated EPCR signaling pathway is cyto-protective under a normal state, and during CM pathogenesis it gets abrogated by sequestered pRBCs. Therefore, restoring the cytoprotective role of EPCR has promising therapeutic potential to successfully treat the CM syndrome in the future. Nonetheless, the role of protective immunity against CM (versus mild malaria) mediated through merozoite surface proteins, e.g., MSP1p19 and MSP3 antibody responses [37], prudently align with the ongoing quest for the new antibody therapeutics for CM. More recently, neutrophil-ICAM-1 (intra-cellular adhesion molecule-1) were identified that could effectively kill pRBCs [38], expressing specific subsets of PfEMP1 variants in CM earlier characterized [39,40]. In addition, it has been recently postulated that a chimeric CIDRα domain of PfEMP1 has immense potential to be a candidate subunit vaccine [41]. These developments in malaria immunology present promising scope and an interface with huge potential to invest in future research efforts directed towards immunotherapeutic exploration in CM. Moreover, a recent clinical trial of IgG antibodies directed against specific PfEMP1 variants (CIDRα1.7 and CIDRα1.8 domains) revealed that they could be acquired early, and specifically ordered in children native to intense malaria endemic regions [42], which is remarkable. It will be worthwhile to monitor the findings from similar trials and forthcoming clinical outcomes from geographically diverse malaria transmission settings. The other important question that could subsequently could arise is whether CM-specific responses in children and adults could be induced by specific PfEMP1-IgG antibodies? With advancements in the field of malaria immunology with respect to neutralizing antibodies specific to the cytoadherence, it will not be unrealistic to anticipate, sooner or later, tailormade, age-specific neutralizing antibody-adjunctive therapies for CM and/or even specific CM vaccines [43,44].

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
