# Peer review of "Pathogenesis of Cerebral Malaria: New Trends and Insights for Developing Adjunctive Therapies"

_pathogens, 2023, doi:10.3390/pathogens12040522_

Round 1

Reviewer 1 Report

This is an opinion paper about future development of adjunctive therapy for cerebral malaria.

The authors present many different aspects of the physiopathology of cerebral malaria up to line 107, including recent findings using novel technologies. In the last paragraph (lines 108-119), the authors propose three areas to be developed: enhancement of EPCR, glutamate antagonist, and transfusion.

The authors should develop the last paragraph (lines 108-119) and provide more information to convince the readers that there are several promising areas for the development of novel adjunct therapy for cerebral malaria. For example, are there studies that succeeded in restoring the role of EPCR? How does glutamate antagonist work to reverse cerebral malaria? The authors propose whole blood transfusion. The literature has many published papers on exchange transfusion in patients presenting hyperparasitaemia. Do the results of these published papers support the authors’ proposal?

English needs a moderate level of correction.

Major comments:

Lines 22-26, “The inability to prevent or treat CM is primarily due to the lack of a specific anti-CM drug or an adjuvant”: In my opinion, this (i.e. “lack of anti-CM drug or adjuvant”) is not the primary reason. We can prevent CM if patients with acute uncomplicated malaria are rapidly diagnosed and treated with existing antimalarial drugs (artemisinin-based combination therapies, ACT) before severe and complicated malaria sets in. Delayed diagnosis and delayed or inappropriate treatment, as well as difficulties in accessing adequate health care, are probably more important factors that explain the inability to prevent CM at present.

Lines 44-47: The authors mention two mechanisms: (i) cytoadherence and (ii) cytokines. What about rosetting?

Lines 58-88, with regards to lines 33-34 “The clinical presentation of CM is diverse in adults and children, and across geographies”: It is known that children and adults do not have the same clinical manifestations of severe and complicated malaria. However, in my opinion, the authors do not provide enough evidence to conclude that there is a geographic difference in the pathophysiology of cerebral malaria between India/Asia and Africa in lines 58-88. Patients in southeast Asia, Western Pacific area, and South American have not been taken into consideration.

Minor comments:

Line 9: cerebral malaria (small letter “c”)

Line 16: delete the comma after “understand”

Lines 24-26, “despite using most potent antimalarials cannot prevent the development of CM”: Please rewrite this sentence with a correct syntax.

Lines 28-30, “CM…Plasmodium falciparum and the human host”: REF 3 is a work based on rodent malaria model and does not support the statement about P. falciparum and the human host.

Lines 39 and 40, REF “Mishra et al 2005”: Are the authors referring to Mishra et al. 2007 (REF 5 in the reference list)? There is no “Mishra et al. 2005.”

Line 43, “absolute mechanisms”: Do the authors mean “pathophysiology”?

Line 46, “resulting immune inflammatory responses leads to…”: lead to…

Lines 44-51, “the major postulations…autopsy-based investigations”: I think that these statements need to be supported by reference citations, unless they are based on the references cited in line 44 (Idro et al. 2010; Rénia et al. 2012).

Line 61, “during the development CM”: during the development of CM

Lines 70-71, “through ADC maps and measuring the elevated plasma levels of lipocalin-2 and miR-150”: through ADC maps and measurement of…

Lines 89-91, “The pursuit of scientific knowledge…”: This sentence is confusing. Please rewrite it with the correct subject of the sentence. What was/were identified?

Lines 95-99, “Presumably so now…clinical scenarios experienced then”: This is not a complete sentence, and the meaning of these statements is not clear. Please rewrite with more clarity, with easily identifiable subject of the sentence and the verb.

Please follow the journal instructions on reference citations.

REF 10 is not correctly cited: Malaria Journal, 6, 138. There is no page number in Malaria Journal. “138” is the article number.

REF 13: Am J Trop Med Hyg 2018, 98, 497–504. Please correct the page numbers.

REF 14: Please delete “a” in superscript.

REF 16: JCI Insight 2021, 6(18), e145823. Please add the article number. Also please delete “b” in superscript.

REF 21: Front Cell Infect Microbiol. 2015, 5, 75. “75” is not the page number. This is the article number.

REF 25: Sci Rep 2021, 11(1), 12077.

Author Response

Reviewer-1

Comments and Suggestions for Authors

This is an opinion paper about future development of adjunctve therapy for cerebral malaria.The authors present many different aspects of the physiopathology of cerebral malaria up to line 107, including recent findings using novel technologies. In the last paragraph (lines 108-119), the authors propose three areas to be developed: enhancement of EPCR, glutamate antagonist, and transfusion.

The authors should develop the last paragraph (lines 108-119) and provide more information to convince the readers that there are several promising areas for the development of novel adjunct therapy for cerebral malaria.

The authors would like to thank the reviewer for the insightful comments and suggestions. The last paragraph particularly has been newly written and developed accordingly, also based on the second reviewer’s suggestion, in the revised manuscript.

For example, are there studies that succeeded in restoring the role of EPCR?

None, to the author’s knowledge and as per published literature. EPCR is a well characterized receptor molecule known for its crucial ability to support the cytoprotective effects of activated protein C (APC) resulting vascular integrity of multiple organ systems including the brain endothelium which gets compromised during cerebral malaria. Thus, the restoration of the role of EPCR is a future direction, as remarked by the authors for this opinion paper.

How does glutamate antagonist work to reverse cerebral malaria?

According to the study by Riggle et al (2018, PNAS), using an experimental model of cerebral malaria and upon treatment with a glutamate antagonist (6-diazo-5-oxo-l-norleucine prodrug, JHU-083), there was reversal of both cytotoxic edema (as evidenced by T2-weighted MRI scans and ADC maps) as well BBB disruption (contrast-enhanced T1-weighted MRI scans).  Earlier to this, Miranda et al (2017, Mol Neurobiol) had investigated the role of another glutamine receptor antagonist (dizocilpine maleate, MK801) in mice model for its neuroprotective role in cerebral malaria. The study could establish the role of MK801 in promoting the balance of TH1/TH2 response and upregulation of neurotrophic factors levels in the frontal cortex and hippocampus, besides partially preventing the abnormalities in the hippocampus as visualized upon MR Imaging.

The authors propose whole blood transfusion. The literature has many published papers on exchange transfusion in patients presenting hyperparasitaemia. Do the results of these published papers support the authors’ proposal?

The authors thank the reviewer for the comment. The authors also agree that exchange transfusion may occasionally present hyperparasitaemia in clinical scenario as per literature.  It may be however, kindly to be noted that the authors do not propose whole transfusion, or any other therapeutic option for cerebral malaria in this opinion paper. As the title of this specific opinion paper, the effort is to highlight the recent trends and emerging insights in the pathogenesis of cerebral malaria which may direct future research to develop new adjunctive therapies for CM.

English needs a moderate level of correction.

All the English corrections, typographies and grammatical errors including re-phrasing of sentences are highlighted in red.

Major comments:

Lines 22-26, “The inability to prevent or treat CM is primarily due to the lack of a specific anti-CM drug or an adjuvant”: In my opinion, this (i.e. “lack of anti-CM drug or adjuvant”) is not the primary reason. We can prevent CM if patients with acute uncomplicated malaria are rapidly diagnosed and treated with existing antimalarial drugs (artemisinin-based combination therapies, ACT) before severe and complicated malaria sets in. Delayed diagnosis and delayed or inappropriate treatment, as well as difficulties in accessing adequate health care, are probably more important factors that explain the inability to prevent CM at present.

The authors thank the reviewer for this comment. We have deleted the the word ‘prevent’ and rephrased the entire paragraph in the revised manuscript.

 Lines 44-47: The authors mention two mechanisms: (i) cytoadherence and (ii) cytokines. What about rosetting?

The authors thank the reviewer for pointing out on rosetting. We have mentioned Rosetting as part of the mechanisms in the revised manuscript.

 Lines 58-88, with regards to lines 33-34 “The clinical presentation of CM is diverse in adults and children, and across geographies”: It is known that children and adults do not have the same clinical manifestations of severe and complicated malaria. However, in my opinion, the authors do not provide enough evidence to conclude that there is a geographic difference in the pathophysiology of cerebral malaria between India/Asia and Africa in lines 58-88.

Patients in southeast Asia, Western Pacific area, and South American have not been taken into consideration.

 The reviewer’s comment is well appreciated. Authors have now taken adequate care to delete the phrases ‘across geographies’. Other phrases and sentences relevant to the differences in the pathophysiology of CM between India/Asia and Africa in lines 58-88. The discussion is kept restricted to comparison only, which was made between Indian and African cohort and based on their recent published findings (Sahu et al 2021, JCI Insight).

Minor comments:

 Line 9: cerebral malaria (small letter “c”)

Corrected.

 Line 16: delete the comma after “understand”

 Corrected.

Lines 24-26, “despite using most potent antimalarials cannot prevent the development of CM”: Please rewrite this sentence with a correct syntax.

Corrected and rewritten.

 Lines 28-30, “CM…Plasmodium falciparum and the human host”: REF 3 is a work based on rodent malaria model and does not support the statement about P. falciparum and the human host.

Plasmodium falciparum is deleted.

 Lines 39 and 40, REF “Mishra et al 2005”: Are the authors referring to Mishra et al. 2007 (REF 5 in the reference list)? There is no “Mishra et al. 2005.”

 The authors thank the reviewer for pointing out this oversight. Necessary corrections have been made.

Line 43, “absolute mechanisms”: Do the authors mean “pathophysiology”?

Yes. Corrected as pathophysiology.

 Line 46, “resulting immune inflammatory responses leads to…”: lead to…

Corrected.

 Lines 44-51, “the major postulations…autopsy-based investigations”: I think that these statements need to be supported by reference citations, unless they are based on the references cited in line 44 (Idro et al. 2010; Rénia et al. 2012).

All relevant literature has been now cited in the revised manuscript.

 Line 61, “during the development CM”: during the development of CM

Corrected.

Lines 70-71, “through ADC maps and measuring the elevated plasma levels of lipocalin-2 and miR-150”: through ADC maps and measurement of…

Corrected.

Lines 89-91, “The pursuit of scientific knowledge…”: This sentence is confusing. Please rewrite it with the correct subject of the sentence. What was/were identified?

Confusing statements are deleted. The entire paragraph is corrected and rewritten.

Lines 95-99, “Presumably so now…clinical scenarios experienced then”: This is not a complete sentence, and the meaning of these statements is not clear. Please rewrite with more clarity, with easily identifiable subject of the sentence and the verb.

The entire paragraph is rewritten with improved clarity.

Please follow the journal instructions on reference citations.

 REF 10 is not correctly cited: Malaria Journal, 6, 138. There is no page number in Malaria Journal. “138” is the article number.

 REF 13: Am J Trop Med Hyg 2018, 98, 497–504. Please correct the page numbers.

 REF 14: Please delete “a” in superscript.

 REF 16: JCI Insight 2021, 6(18), e145823. Please add the article number. Also please delete “b” in superscript.

 REF 21: Front Cell Infect Microbiol. 2015, 5, 75. “75” is not the page number. This is the article number.

 REF 25: Sci Rep 2021, 11(1), 12077.

 All the references have been formatted as per the reviewer’s suggestions and according to the journal instructions.

Reviewer 2 Report

Authors are eminent researchers in the field of Cerebral malaria pathology, The review article is well described masterpiece of recent trends in cerebral malaria in respect adjunct therapies and future directions.

I have few suggestions

A) In last couple of years there were advancement of metabolites and neutralizing antibodies in the field of cerebral malaria, If authors can add their expert opinion and guide readers in this new fields of research will be useful of wider audiences.

B) References in the manuscript are in alphabetical order, reference lists are numbered though. 

Author Response

Reviewer-2

Comments and Suggestions for Authors

Authors are eminent researchers in the field of Cerebral malaria pathology, The review article is well described masterpiece of recent trends in cerebral malaria in respect adjunct therapies and future directions.

I have few suggestions

  1. A) In last couple of years there were advancement of metabolites and neutralizing antibodies in the field of cerebral malaria, If authors can add their expert opinion and guide readers in this new fields of research will be useful of wider audiences.

The authors would like to gratefully acknowledge the reviewer for the kind words of appreciation. Indeed, it was an extremely constructive suggestion by the reviewer which directed the authors to develop the a new (last) paragraph in the revised manuscript, particularly based on the recent advancement in neutralizing antibodies specific to CM and important insights on the protective immunity against CM, including a most recent study proposing a candidate subunit vaccine for malaria.

  1. B) References in the manuscript are in alphabetical order, reference lists are numbered though. 

As suggested, the reference list is now corrected to be in alphabetical order.

Round 2

Reviewer 1 Report

The authors have satisfactorily revised their opinion paper and further updated the cited references.

MINOR COMMENTS:

Line 48: pathophysiology leading to the development of CM and brain injury [delete comma here] remains [present tense] an enigma

Line 51: cytoadherence of the pRBCs to the endothelial cells, facilitated by rosette formation in the cerebral microcapillaries [insert a comma here] causes…

Line 56: The breach in the BBB causes brain swelling, which manifests [singular verb in agreement with “swelling”] as…

Line 58: delete the comma after “that”

Line 60: mannitol (small letter “m”)

Line 61: insert a comma after “edema”

Line 62: insert the references (Namutangula et al 2007; Mohanty et al 2011) at the end of the sentence, after “beneficial effects”

Line 65: delete the comma after “degrees”

Line 69: delete the comma after “nuero-radiological studies”

Line 71: pathogenetic mechanisms of action (instead of “mechanisms in action”?)

Line 93: delete (Sahu et al 2021b). The authors start this sentence with the phrase “in this study” referring to Sahu et al. 2021 in the preceding sentence (lines 85-91). There is no need to repeat “(Sahu et al 2021b)” in line 93.

Line 94: was found to be consistent with

Line 102: insert a comma after “severe and cerebral malaria”

Line 105: clinical trials [plural]…have yielded

Line 109: The recent series of studies

Line 111: delete the comma after “determinants”

Line 120: delete the comma after “emerging”

Line 121: will become increasingly important

Line 122: should be monitored

Lines 123-129: the role of glutamate antagonists… and (not “or”) whole blood transfusion therapy…have exhibited

Lines 146-147: CM [delete the comma] characterized earlier or in previous studies (Lennartz et al. 2017; Adams et al 2020)

Line 160: it will (not “won’t”) not be

References: Contrary to what the authors state in their replies to reviewers, they did not follow the journal instructions. References are numbered in the text in the order of citation. The list of references at the end of the manuscript should also be presented in the order of citation, and not in alphabetical order of the first authors.

Author Response

Kindly refer to the file attached.
